# Post-Earthquake Assessment of a Historical Masonry Building after the Zagreb Earthquake–Case Study

**Ivan Hafner, Damir Lazarević, Tomislav Kišiček * and Mislav Stepinac**

Faculty of Civil Engineering, University of Zagreb, 10000 Zagreb, Croatia;
ivan.hafner@grad.unizg.hr (I.H.); damir.lazarevic@grad.unizg.hr (D.L.);
mislav.stepinac@grad.unizg.hr (M.S.)
* Correspondence: tomislav.kisicek@grad.unizg.hr

**Abstract:** After the Zagreb earthquakes in March of 2020, around 25,000 buildings were estimated damaged, most of them being in the historic city center. This fact is not that surprising since most of the city center buildings are unreinforced masonry structures that have not been assessed in quite some time and usually no retrofitting methods were ever applied. The rapid post-disaster assessment began the same day after the first earthquake occurred. Through mostly visual assessment methods, the basic idea is to identify the safety and usability of buildings in general. This type of assessment was also conducted in one of the oldest Croatian cultural institutions, Matica Hrvatska. It is a building of great historical significance and cultural value, as is most of the city center. Accordingly, this building was constructed with no consideration given to seismic events and with the use of traditional materials and building techniques. In the scope of this paper, urgent actions that were taken are shown with problems and challenges that occurred. Furthermore, the decision-making process after an earthquake is elaborated. In addition, a numerical model is developed in 3Muri software for structural modeling. A non-linear static pushover analysis is performed, and possible failure mechanisms are examined. Furthermore, real-life damage is compared to the software results, and a conclusion process of the building's usability is explained. In the end, the results obtained are analyzed and conclusions regarding the efficiency of the used software are drawn.

**Keywords:** assessment; earthquake; unreinforced masonry; case study; cultural heritage; Zagreb

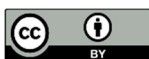

## 1. Introduction

On 22 March 2020 an earthquake of medium magnitude $M_L = 5.5$ and intensity of VII hit the Zagreb Metropolitan area, with the epicenter being just 7 km North of the city center [1]. The earthquake was 10 km deep which puts it in the category of shallow earthquakes. The recording of the peak ground acceleration (PGA) for the Zagreb earthquake is shown in Figure 1 [2]. Besides the mostly residential buildings in the vicinity of the earthquake's epicenter, most of the damaged buildings were located in the Lower and Upper town areas in the center of the city. The damaged buildings, mostly unreinforced masonry structures, include residential buildings, schools, universities, public buildings and even hospitals. Since most of these buildings were designed and built before the development of any seismic regulations this fact is not that surprising. It should also be considered that the vast majority of residential buildings and even some schools and cultural buildings were not assessed in quite some time and were never retrofitted. Consequently, the damage to these buildings was significant (Figure 2). In Europe, the popularity of unreinforced masonry buildings in the built stock is very wide due to a continuous tradition of this construction technique. Croatia is not an exception in this context, and particularly the city of Zagreb, where numerous masonry buildings can be found. An additional problem rises from the fact that most of these buildings in the city of Zagreb are parts of city blocks (aggregate buildings). Masonry building aggregates constitute a significant part of

the historic European cities. Due to population growth and city expansions in the 19th century, building blocks were formed as an appropriate solution. This structural form is very common all over Europe, and its architectural and cultural values cannot be neglected. For instance, Eixample district is one of the most recognizable landmarks of Barcelona and the whole Lower Town district in Zagreb is under cultural heritage protection. Hence, there is an ever-growing need for a comprehensive, protective and sustainable approach to such complex issues. For civil engineers these structures represent a special challenge as there is a difference in the seismic behavior of buildings attached to a block and those that are isolated. It is acknowledged that the seismic assessment of a single building can be investigated using well-defined methodologies; however, changing the scale of interest from a single building to a whole aggregate, the complexity of problems increases considerably. One additional problem is that the relevant codes do not provide detailed guidance on the building blocks. Due to the complex behavior of block buildings and large number of users which implies a higher risk, it is crucial to assess safety and seismic vulnerability [3].

The case study building described in this paper is a corner building of a city block. Behavior of such buildings is quite complex because of the affect that the adjacent buildings have on the building observed. The role of interaction between adjacent buildings during seismic events has been investigated by many authors [3–7], however, due to the complexity of the problem, its solution is not always straightforward. Therefore, the crucial part of strategies for seismic risk mitigation and for improving the resiliency of cities should be the continuous vulnerability assessment [8]. Although, seismic vulnerability assessment of historical masonry buildings in Croatia was conducted for the Adriatic coast [9], that is not the case for the city of Zagreb which is located in the continental part of the country. However, with the lessons learned from the Zagreb earthquakes, it is obvious that the assessment and rehabilitation of existing masonry structures must be performed at the highest level using a variety of state-of-the-art techniques. When talking about the assessment of existing masonry buildings per se, the variety of techniques enables the civil engineering community to have a very good insight into the building's characteristics and behavior. From laser scanning, use of unmanned aerial vehicles (UAVs) and LiDAR (Light Detection and Ranging) to numerous NDT (nondestructive testing) methods [10–13] and the use of machine-learning based vulnerability analysis [14] or automated mechanical models for masonry aggregates [15], all may be used to get a better understanding of the buildings' most important characteristics.

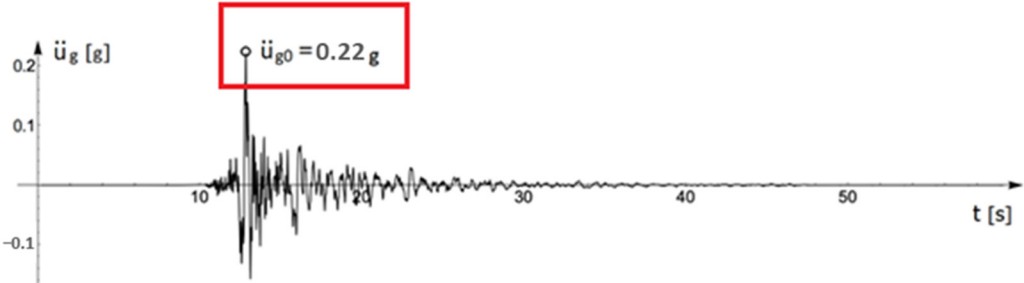

**Figure 1.** The recording of the peak ground acceleration (PGA) for the Zagreb earthquake in the dominant North–South direction [2].

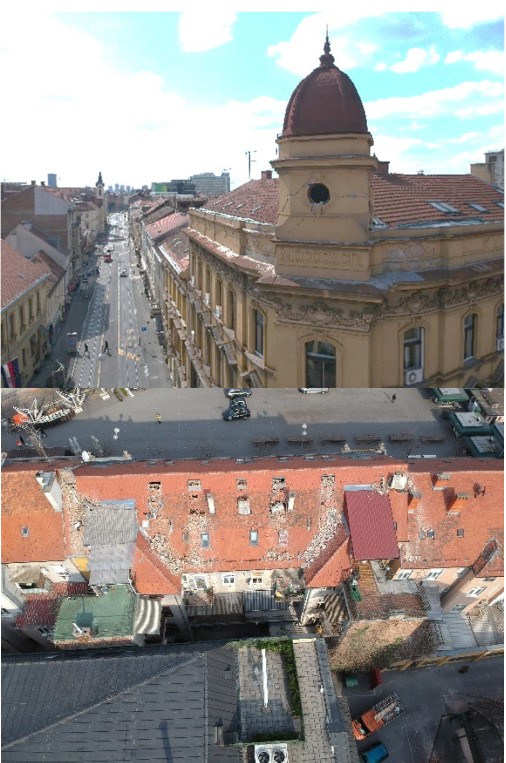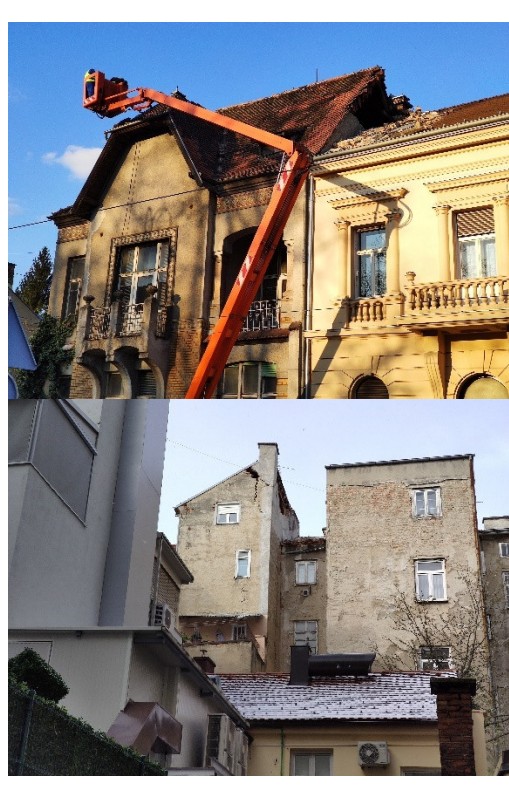

**Figure 2.** Typical damage of buildings in the city center of Zagreb (photo credit: Mislav Stepinac).

The rapid post-disaster assessment began the same day as the first earthquake. Through mostly visual inspection, the usability of damaged buildings is assessed. The priority of this assessment is to determine if the building is usable, temporarily unusable, or completely unusable regarding the protection of human life and surrounding property. For this purpose, every building was assigned a usability label. Therefore, a building can be categorized as usable (green label), temporarily unusable (yellow label) and unusable (red label). Similar types of assessment methods have been used across the world [16–20]. The full explanation of the usability levels is given in [21] and Appendix A. As previously mentioned, the urban center of the city of Zagreb consists of mainly unreinforced masonry buildings. The buildings are characterized by densely built, usually rectangular blocks of buildings made of stone, brick or a combination of these materials. These buildings are composed of massive orthogonal or longitudinal walls with ceilings built as masonry vaults or structures composed of wooden beams. The roofs are, as well, usually wooden.

According to [1], 72% of buildings in the historical center of the city suffered significant damage due to the earthquakes in 2020 according to preliminary assessment results. The majority of the buildings in the city center are residential buildings followed by cultural institutions and state and government buildings and are, more or less, under heritage protection. The total number of the earthquake-affected non-residential buildings under heritage protection comprises 192 cultural institutions, 13 state government heritage buildings, and 159 sacral religious buildings. Since most of the buildings in the sector are very old masonry buildings with load-bearing walls, columns, arches, and vaults of various shapes, moderate to severe structural damage was suffered by 118 buildings (32%), and heavy structural damage was reported in 41 buildings (11%). The total damage according to the World Bank report to buildings and other physical assets in cultural sector is estimated at 1.38 billion EUR [22]. Figure 3 shows the number of damaged heritage buildings and the total financial damage according to the World Bank report.

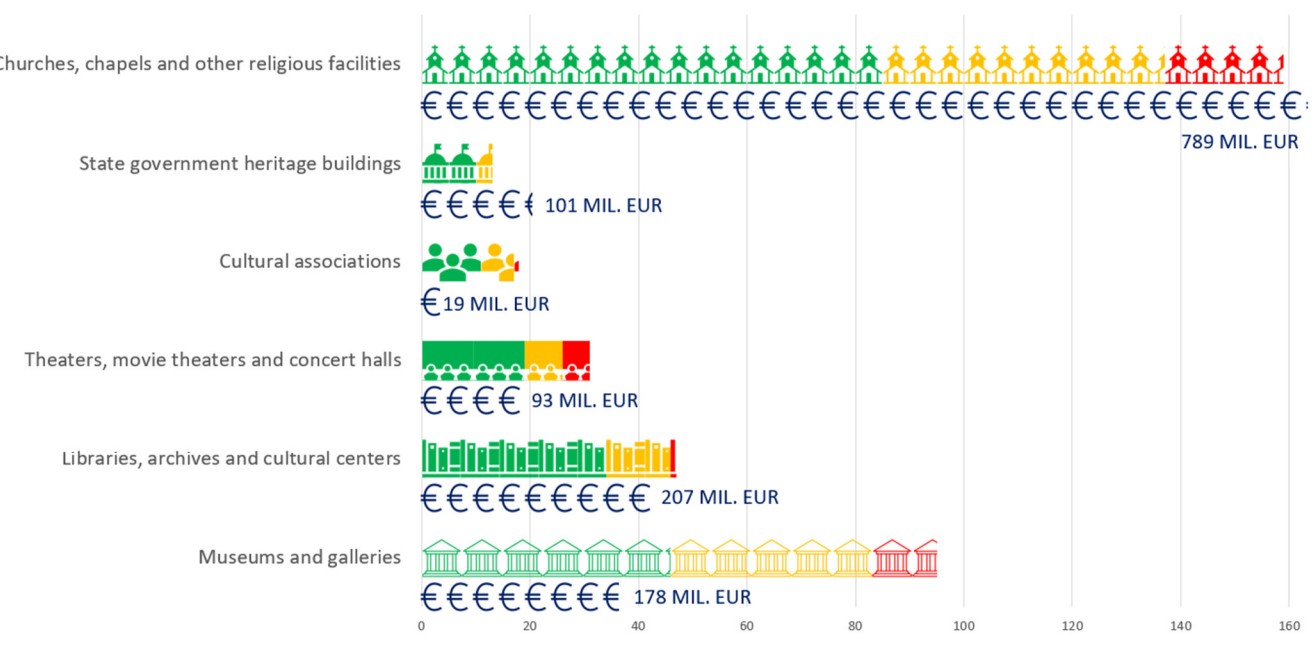

**Figure 3.** Damage to the heritage buildings after the Zagreb earthquake in 2020.

This paper focuses on the rapid post-earthquake assessment of a typical masonry building in the city of Zagreb. After the introduction, the case study building will be presented with the description of the load bearing elements and materials used for its' construction. The case study building is under the protection of the Ministry of Culture, the Directorate for the Protection of Cultural Heritage, and the City Institute for the protection of Cultural and Natural Monuments in Zagreb. The reason for choosing this building lies in the fact that in terms of geometry, material and method of construction, it is a typical building from the 19th century, which are dominant in the city center. In the following section the rapid post-earthquake assessment will be explained in detail and the damage obtained will be presented. At the very end of the section the decision-making process regarding the usability of the building will be explained. A numerical analysis is conducted in the fourth section. A more refined non-linear static or pushover analysis is recommended and adopted for existing buildings such as this one [23]. The main problems of pushover modeling for such buildings are addressed. In the manuscript, an additional goal is to understand if this method can be safely applied for corner buildings with an asymmetric floorplan. The model shows just an indication of the structural behavior of the case study building. In the conclusion section, various points will be raised about the validity of this method in such buildings, especially those that have wooden beam floors.

### 2. The Case Study Building

In this paper, a building located at Matice Hrvatske 2 Street in Zagreb will be examined. As an institution, Matica Hrvatska, was founded in 1842 with the main objective of promoting the national and cultural identity of Croatia with an emphasis on the artistic, scientific and economic milieu. It is considered to be one of the largest and most prominent book and magazine publishers in Croatia. Besides publishing, Matica Hrvatska acts as an organizing body for a great number of cultural and scientific events such as book presentations, scientific symposiums and lectures, panels and even classical music concerts. In the scope of this organization is also the promotion of young artists, musicians, lecturers and prospective students in a wide range of fields. From a cultural, historical and scientific perspective it is one of the most important buildings in Zagreb and in Croatia [24].

From a structural standpoint, it is an L-shaped building situated at the Northwest corner of a city block in the city center (Figures 4–6). This Neo-Renaissance building was

built in 1887, with the Western annex of the building finished in 1892. The final shape of the building was reached in 1921 with the construction of the Northern wing annex. From that point forward, two adaptations of the building were carried out. In the first adaptation in 1978, the ceiling at the highest floor (attic floor) was retrofitted using steel beams. In today's standards, a steel beam would be considered IPE240, and it is connected to columns with similar dimensions. The roof was also refurbished. Moreover, in 1993 a smaller reconstruction of the basement was carried out. The building's floor dimensions are 14 m × 42 m for the Western wing (North–South direction) and 16 m × 28 m for the Northern wing (East–West direction). There is also another smaller annex at the South side of the Northern wing, with the floor dimensions being 10 m × 7 m. In total, the floor area of the building is approximately 890 m². In the vertical direction, the building has a basement, three floors and an attic. The height of the floors varies from 4.2 m to 4.8 m, with the second floor being the highest. The attic is 5.0 m high.

As for the materials used, the entire building is built in solid brick of an old standard format 29 cm × 14 cm × 6.5 cm and with lime mortar which were used in the late 19th and at the beginning of the 20th century. Accordingly, the load-bearing walls are 30, 45 and 60 cm thick. The wall thickness varies throughout the building, with the thickest walls found on the lowest floor and the thinnest on the third floor and the attic. The plaster thickness is usually 3 cm throughout the building. The ceiling structures are constructed out of wooden beams with a cross-section of 20 cm × 20 cm placed 0.80 m apart with a 2.4 cm thick single wooden plank over them. This type of floor carries the load in one direction. As it has already been mentioned, the attic floor was reinforced with the use of steel beams that are placed 2.40 m apart. The dimensions of the openings also vary throughout the building. The width of the openings is constant throughout the building, 1.15 m for smaller windows and 1.4 m for larger ones. The height of the openings varies from 2.1 m to 2.6 m with the highest windows being the ones at the second floor.

The case study building is under heritage protection and special assessment techniques should be applied. Cultural heritage is still not receiving sufficient consideration in disaster risk management planning, and the lack of integration of cultural heritage protection measures into national, regional or local risk management strategies [25] was insufficient in the case of Zagreb, and particularly this building.

As the building is under heritage protection, special rules and integrated geometric survey for the conservation should be applied [26]. Similar to the paper by Milić et al. [27] a detailed survey of the external dimensions and façade was created to preserve its architectural value. The aerial view of the building is shown in Figures 4 and 5. Laser scanning was performed with the Leica BLK360 device and processed in the Cyclone Register 360 software (Figure 6).

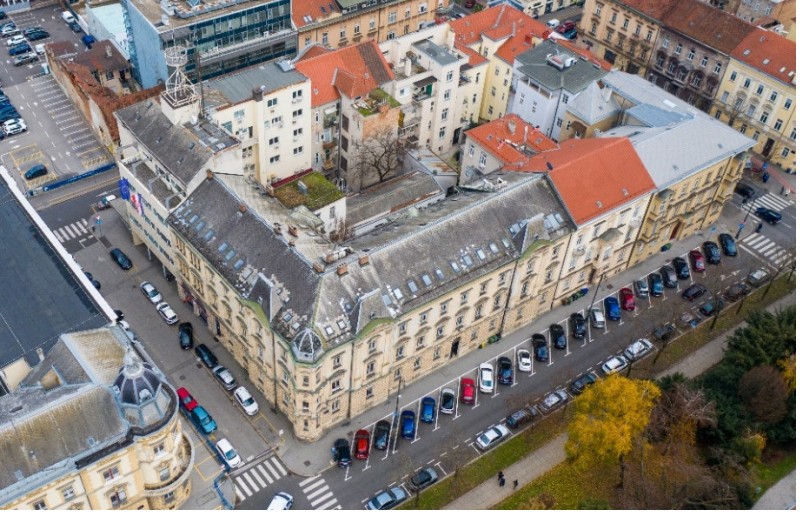

**Figure 4.** Aerial view of the building (photo credit: Mislav Stepinac).

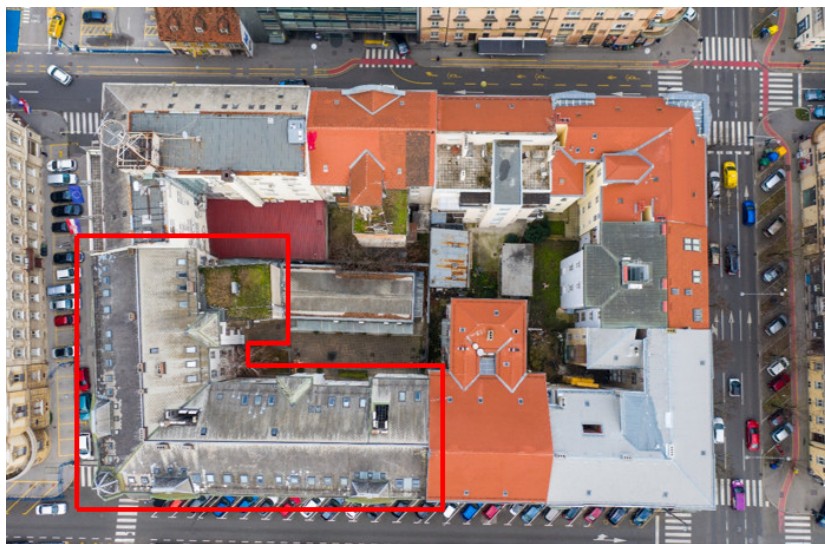

**Figure 5.** Position of the case study building inside of the city block (photo credit: Mislav Stepinac).

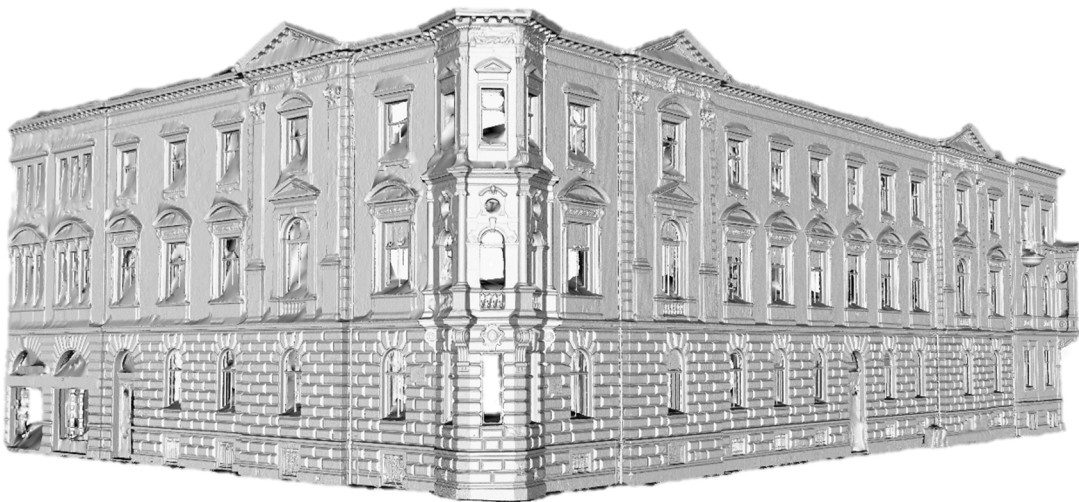

**Figure 6.** Laser scan of the building's façade.

### 3. Rapid Post-Earthquake Assessment of the Building

After an earthquake or any other hazardous event, the first step in the assessment process is a preliminary inspection of a building. This preliminary inspection is conducted in several steps. The methodology of the assessment process is defined as follows:

- Inspection of the available documentation and archives regarding the building's construction history and geometry
- Damage assessment based on a visual inspection—damaged load-bearing elements and damaged non-structural elements that pose a threat to human lives or to the surrounding property
- Damage assessment based on a visual inspection—identification of different types of damage (crack patterns, local plaster damage, etc.) and the development of damage schemes for all structural elements of the building
- Information gathering regarding materials quality—assessment of the quality of mortar, condition and type of masonry elements and the condition of timber used in the floor structures (humidity, density, etc.)
- Data collection regarding the dimensions of certain elements and important measures (cross section of wooden beams, distance between wooden beams,

masonry elements dimensions, mortar thickness, etc.)—very important for the numerical modelling of the building–defined in Section 2 for this case study

- The decision-making process—the appointment of one out of three usability labels explained in Appendix A

The main goal of this assessment is to check the usability of the building. The rapid assessment is conducted as soon as possible after a seismic event, bearing in mind the safety of civil engineers conducting the investigation. Additionally, state-of-the-art technologies may be used to understand better the building's usability and condition after a visual inspection. For a better understanding of the building's exterior state of damage, laser scanning was performed with the Leica BLK360 device (Figure 4) and an UAV was used (Figures 5 and 6). Additionally, laser scanning or drone imaging may form a point cloud of the exterior and the interior of the building [9]. In the end, if it is possible, non-destructive testing (NDT) methods should be used [28–31]. The use of such methods is very helpful in the numerical modeling process.

On the 15 April 2020, an inspection of this paper's case study was conducted. The main goal of the assessment was to investigate the extent and type of damage to the structural and non-structural elements in the building. The building was given a yellow label which meant it was temporarily unusable. The damage found in the building may be placed in the following categories:

- Different crack patterns on the ceilings and above the openings (Figure 7a,b)
- Longitudinal cracks in joints between the walls and ceilings
- Diagonal X-pattern cracks on parapet walls between the openings on the façade (Figure 8)
- Cracks on load-bearing walls and cracks on walls where previous interventions were made, such as bricked-up doors (Figure 9a,b)
- Smaller cracks in the staircase of the building

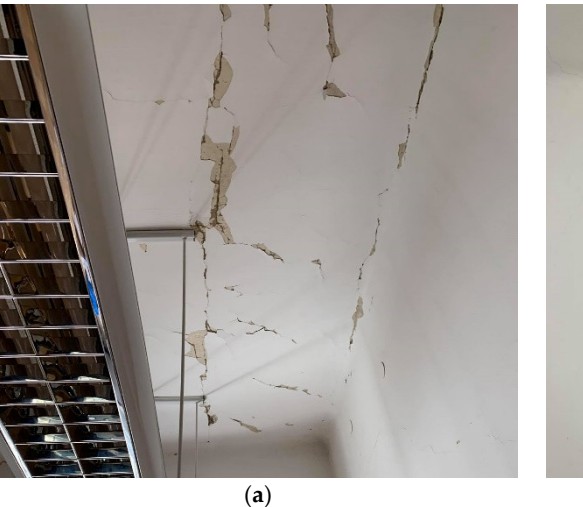 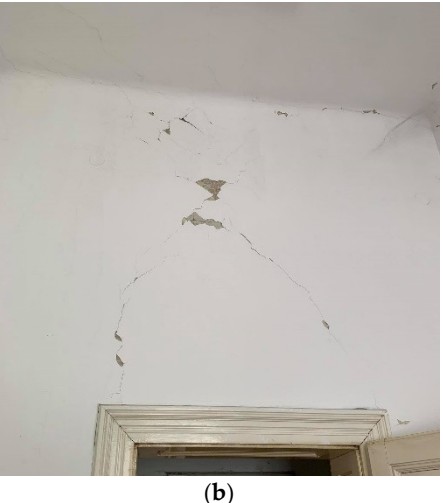

(**a**)                (**b**)

**Figure 7.** Cracks in the ceiling (**a**) and above the openings (**b**) on the 2nd floor.

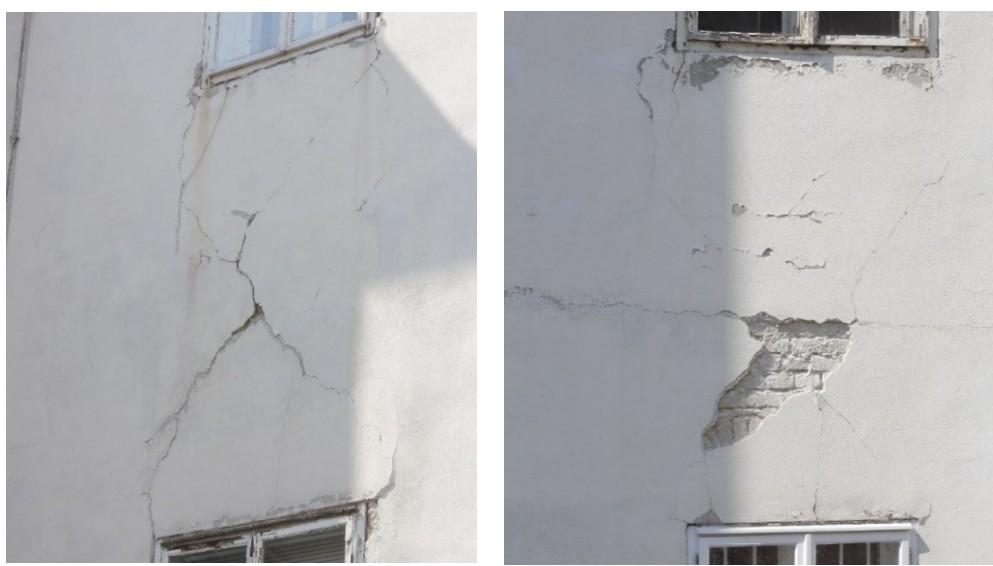

**Figure 8.** Diagonal X-pattern cracks on parapet walls between the first and second floor.

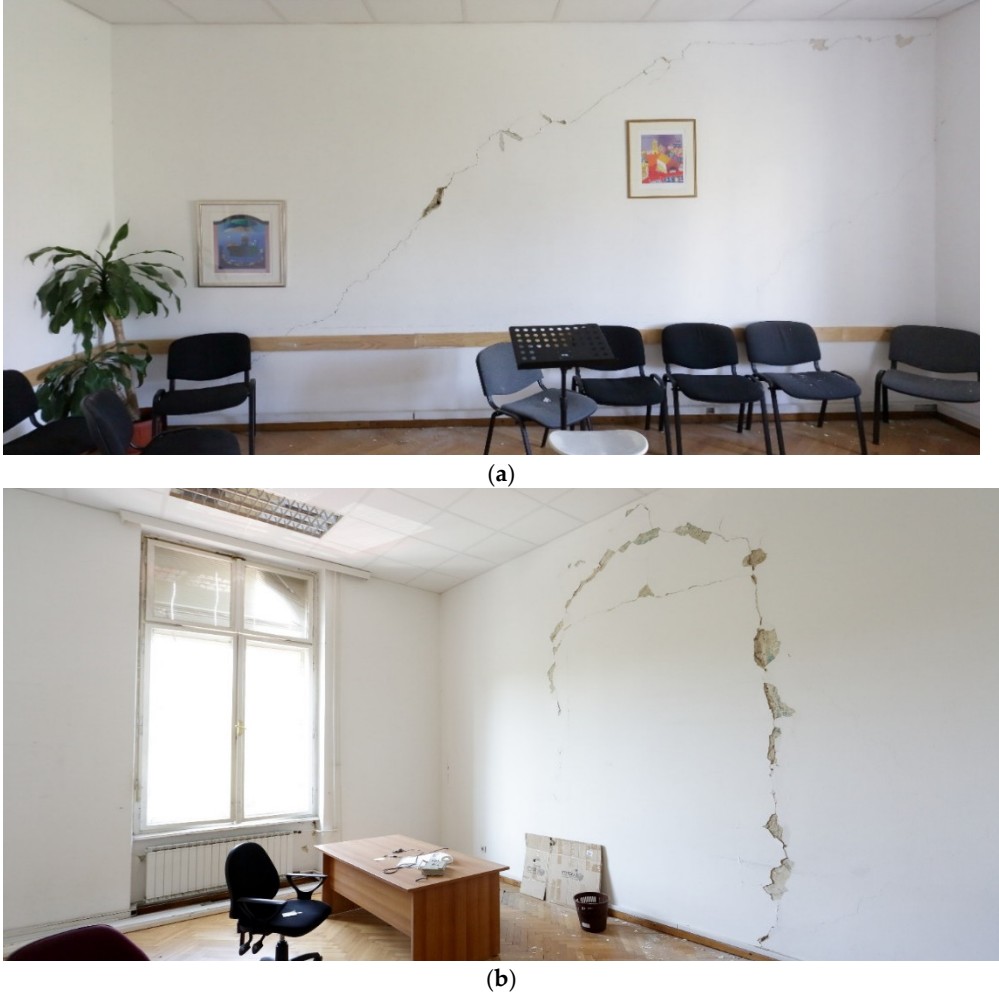

**Figure 9.** Cracks on load-bearing walls (**a**) and cracks on walls where previous interventions were made such as bricked-up doors and above the openings (**b**) on the 3rd floor.

All damage that is found in the building is photographed and described. Besides that, every type of damage is plotted in the building's 3D schemes according to their severity

(Figures 10–13). The green color (smaller damage) represents all types of damage where there is no need for a detailed check-up. Small local damage of plaster, local cracking on walls where the openings were bricked up, very thin cracks etc., fall into this category. The yellow color (medium damage) represents all types of damage where the plaster should be removed, and a detailed check-up is needed such as for the diagonal cracking of load bearing walls and severe cracks on parapet walls. In the end, the red color (severe damage) represents severe damage of structural elements where retrofitting methods must be used. This way, how the building behaved during the earthquake may be obtained.

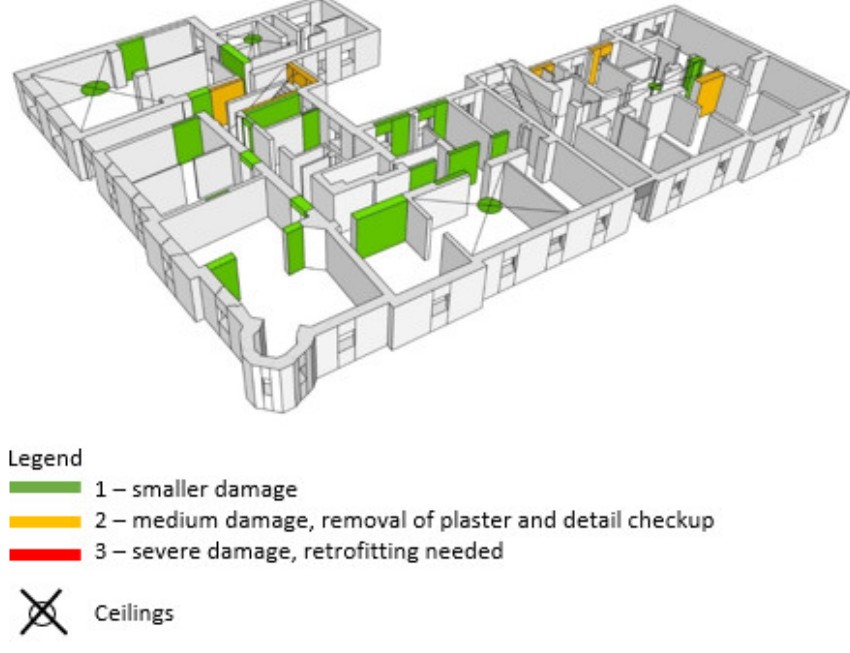

Legend
1 – smaller damage
2 – medium damage, removal of plaster and detail checkup
3 – severe damage, retrofitting needed

Ceilings

**Figure 10.** Damage scheme for the 1st floor of the building.

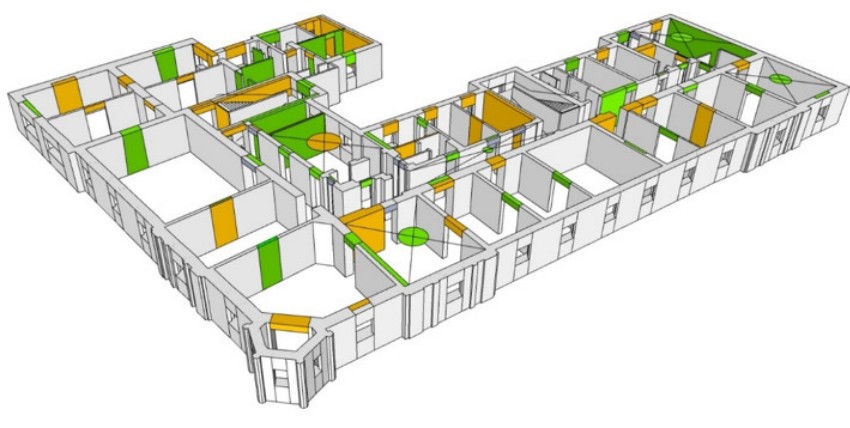

Legend
1 – smaller damage
2 – medium damage, removal of plaster and detail checkup
3 – severe damage, retrofitting needed

Ceilings

**Figure 11.** Damage scheme for the 2nd floor of the building.

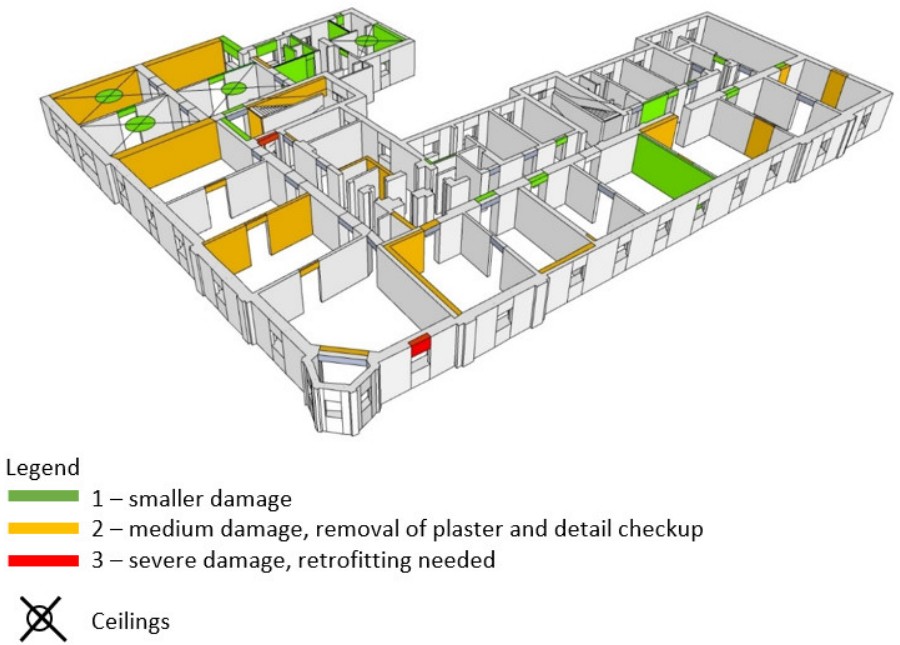

Legend
- 1 – smaller damage
- 2 – medium damage, removal of plaster and detail checkup
- 3 – severe damage, retrofitting needed

⊗ Ceilings

**Figure 12.** Damage scheme for the 3rd floor of the building.

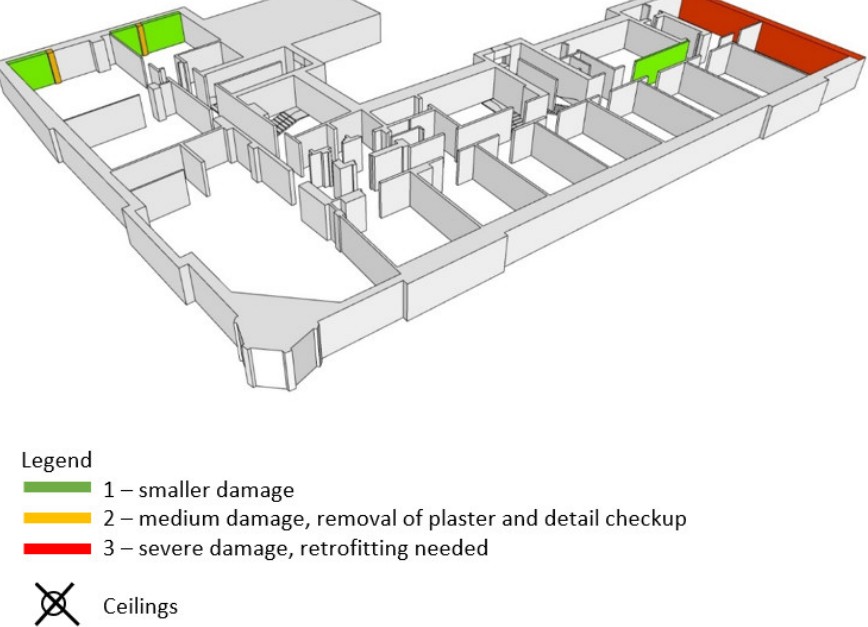

Legend
- 1 – smaller damage
- 2 – medium damage, removal of plaster and detail checkup
- 3 – severe damage, retrofitting needed

⊗ Ceilings

**Figure 13.** Damage scheme for the attic of the building.

After the inspection, the following damage was recorded:

- The basement: no damage was found that could be linked to the earthquake.
- 1st floor/ground floor: different types of damage were revealed. As it can be seen in Figure 10, no severe damage was found on the ground floor. The majority of the damage is visible in the form of light cracks or local decay of plaster on the wall coverings, vaults and ceilings. Moreover, minor diagonal and X-pattern cracks are visible in the walls over the doors and parapets between the windows on the façade. A few more damaged parts of the building's ground floor include the Eastern staircase and the load-bearing walls in the South–North direction located in the most Southern part of the building (Figure 10). Again, the damage is mostly visible in the form of

diagonal cracks. In this particular case the removal of plaster was needed with a more detailed inspection to be conducted.

- 2nd floor: more damage was detected than on the first floor, as can be concluded from Figure 11. The damage pattern is quite similar, with most of the damage being cracks and plaster decay. More severe damage was detected in wooden ceilings and load-bearing walls. Most of the cracks are diagonal or in an X-pattern, with some being longitudinal across the wall. These longitudinal cracks are typical for locations where there is a difference in the materials used or where electrical wiring is placed. Another type of cracking is found on the second floor. In the locations where, throughout the years, openings or doors were closed, excessive damage may be found (Figure 11). These types of cracks appear due to the differences in material used for the construction of the wall and the material used for bricking up a door or an opening in the same wall. This type of damage is very typical in old masonry buildings. Most of the damaged parts of the second floor need to be inspected with more detail.
- 3rd floor: first signs of severe damage may be found. As shown in Figure 12, the red color indicates that retrofitting methods must be used. In particular, a load-bearing wall at the North-Western corner of the building and a wall in the Eastern part of the building were severely damaged. Besides that, on the third floor, the diagonal and X-pattern cracks are in some cases quite wider than on the first and second floors. Furthermore, the cracks extend throughout the length of the wall, which means that the entire wall has fulfilled its purpose when it comes to its shear strength. A detailed inspection is needed for the vast majority of the damage found on the third floor.
- Finally, in the attic, no specific damage was found except the Southern gable wall, which practically needs to be rebuilt and is a danger to any bystander in the street. A more detailed assessment is needed in the Western gable wall (Figure 13).

The quality of mortar via visual check and the condition and type of masonry elements were checked throughout the structure. The masonry used is of good quality and is in adequate condition. The mortar on the other hand is in quite bad shape. It can be easily removed by hand and at some locations there was practically no mortar in the bed joints. The condition of timber used in the floor structures was checked in a couple of locations. The timber used is of adequate quality and the connections with the masonry walls are satisfactory.

In conclusion of the rapid assessment, the building of Matica Hrvatska was categorized as temporarily unusable due to the determined damage, falling plaster and overall employee safety. However, the load-bearing capacity of the building is not impaired. Therefore, after the debris is cleaned and a more detailed inspection of certain parts is conducted, the building may be used safely.

## 4. Numerical Model–Pushover Analysis

The 3D numerical model was created in 3Muri12 software (developed by S.T.A. Data, Torino, Italy) [32]. The software is based on a macro-element (equivalent frame model) approach and was chosen due to its computational efficiency and adequate precision [33]. The macro-elements (or non-linear beam elements) used in 3Muri12 software are divided into three categories: piers, spandrels and rigid nodes. All deformations are concentrated in piers and spandrels, and they are connected with rigid nodes. For the use of the software, official user manuals were used with a variety of useful tutorials and similar journal articles dealing with such case studies [18,34–38]. The 3D model of the building and the equivalent frame made of macro-elements is shown in Figure 14.

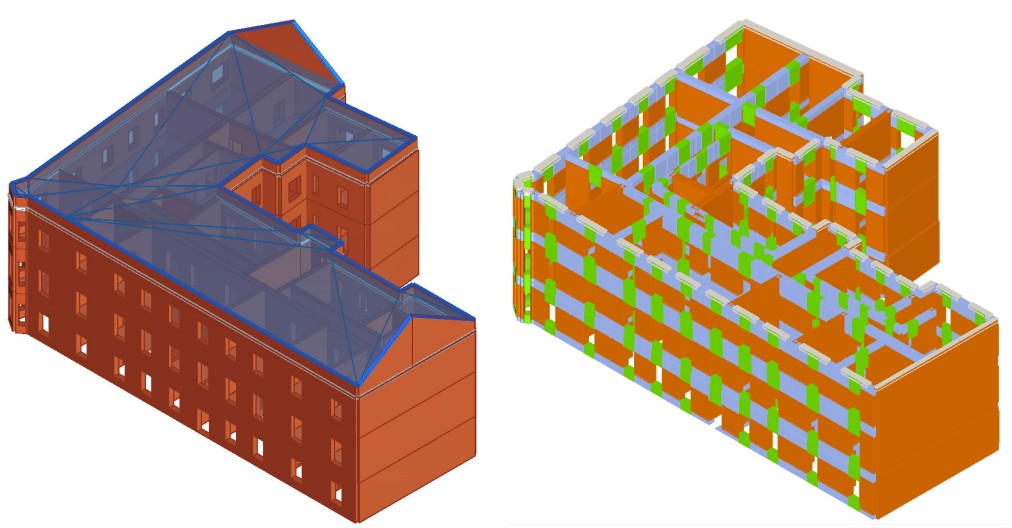

**Figure 14.** The 3D model and the 3D equivalent frame of the building in 3Muri12 software.

In 3Muri12 software, a non-linear static pushover analysis is used. This method allows a comprehensive insight into the critical elements and sections of the building, local failure mechanisms that may occur and the global behavior of the building as a whole. The basic principle of this method is that the vibrations of the building are assessed only in the two translational directions. The analysis is performed with constant gravitational loads and a monotonous increase of the horizontal loads. When it comes to the horizontal loads, two distributions are used in the pushover analysis. The first distribution is represented by a uniform pattern across the building's height. The second one uses a modal pattern distribution where the horizontal load is distributed along with the building's height proportionally to the first vibration mode shape of the building determined through the elastic analysis [39,40]. Overall, there are three main failure modes of masonry walls [41–43]. In unreinforced masonry structures, the usually dominant failure mode is the diagonal cracking failure [44]. This type of failure is known as the Turnsek–Cacovic constitutive law. In 3Muri12 software, the dominant failure mechanism is chosen in the material properties. For this case study building, the Turnsek–Cacovic law is chosen.

Even though there are other, more refined methods for the assessment of existing masonry buildings [45], the beam-based macro-element modeling approach used in 3Muri12 gives more than satisfactory results. Additionally, this simplified approach is less demanding and more practical to apply in engineering practice [46]. The main influence of the vertical component of a seismic action (z-component) was not considered in this study, although it might have an influence on a global behavior of a building [47,48]. In Croatia it is the most used approach which is the biggest reason why it was chosen for this research.

### 4.1. Material Characteristics and Other Input Data

Since there were no experimental in situ tests conducted, the uncertainties in the assessment phase were covered by mean values of the material characteristics taken according to the working draft for the new Eurocode EN1998-3 [49]. Uncertainties defined by Tomić et al. [50] like the non-linear connections, such as wall-to-wall and floor-to-wall connections were not taken into account. In this document, the mean values of material characteristics are given for different masonry types. For solid brick masonry and lime mortar the values are shown in Table 1 [50]. Besides the material characteristics, a few other important factors are used as input data in the software. The confidence factor (FC) is an indicator of the achieved knowledge level. Since there were no in situ tests conducted, but the inspection of the building was quite detailed, the authors of this paper

decided to take the confidence factor (FC) as 1.2 (knowledge level 2–normal knowledge). The partial safety factor for the material used is taken following Eurocode EN1996 [51]. The values of the maximum shear drift for the shear failure and the maximum bending drift for the bending failure are taken as 0.004 and 0.008, respectively [23,37]. The final creep coefficient was taken according to Eurocode EN1996 [51]. Besides the aforementioned properties, the properties of the wooden floors had to be taken into account using subjective engineering judgment. In the 3Muri12 software, they were defined as one-way timber floors with a single wood plank over them. The dimensions were taken as it was already defined in Section 2. The usual timber used in this building type is softwood, spruce, or pine. The C20 class according to EN1194 [52] was adopted. For that strength class the mean modulus of elasticity may be taken $E_{0,mean}$ = 9500 N/mm².

**Table 1.** Mean values for mechanical properties of solid brick masonry and lime mortar.

| Mechanical Property | Value |
| --- | --- |
| Modulus of normal elasticity (E) | 1500 N/mm² |
| Shear modulus (G) | 500 N/mm² |
| Unit weight of masonry (w) | 18 kN/m³ |
| Mean compressive strength of masonry ($f_m$) | 3.4 N/mm² |
| Initial shear strength of masonry ($f_{v0}$) | 0.160 N/mm² |
| Confidence factor (FC) | 1.2 |
| Partial safety factor for material | 1.5 |
| Shear drift | 0.004 |
| Bending drift | 0.008 |
| Final creep coefficient | 0.5 |

*4.2. Static and Seismic Analysis*

The first step is to perform the static analysis according to [51,53,54]. Then, the static loads are calculated manually and inserted as surface loads on the timber floors. After the static loads were applied, the seismic analysis was performed.

After the Zagreb earthquake, the Croatian parliament passed a new legislation, "Law on the Reconstruction of Earthquake-Damaged Buildings in the City of Zagreb, Krapina-Zagorje County and Zagreb County (NN 102/2020)" [55]. According to this law, the return period of an earthquake varies depending on the level of strengthening for old masonry buildings damaged in recent earthquakes. The limit state of significant damage (SD) with a return period of 475 and limit state of damage limitation (DL) with a return period of 95 years were checked. According to the Croatian seismic hazard map, the belonging peak ground acceleration intensities are $a_{gR}$ = 0.251 g (SD) and $a_{gR}$ = 0.126 g (DL), respectively for that location [56]. Furthermore, in the new law, the return period of 225 years which corresponds to a probability of exceedance of 20% in 50 years, is introduced for a limit state of significant damage (SD). The near-collapse (NC) limit state is not included in the law and is not used in the software accordingly. The soil type in the city of Zagreb may be taken as soil type C. For the seismic load input, an importance factor needs to be chosen. The Matica Hrvatska building belongs to importance class III [57]. Therefore, the importance factor is taken as $\gamma_I$ = 1.2.

Furthermore, in the model itself, a couple of simplifications were made. Firstly, the horizontal loads for the pushover analysis need to be applied at the location of the center of the masses at each floor level. Since the floor plan of the building is very complex, the location of the center of the masses was approximated. The position of this node on the top floor is shown in Figure 15. Additionally, because of the complex floor plan, the roof was not modeled. Fortunately, in the seismic analysis the roof may be excluded as a load-bearing structure since it does not significantly affect the structure's seismic response and does not contribute to its resistance. The contribution of the roof for the static and seismic analysis was considered a static load on the structure.

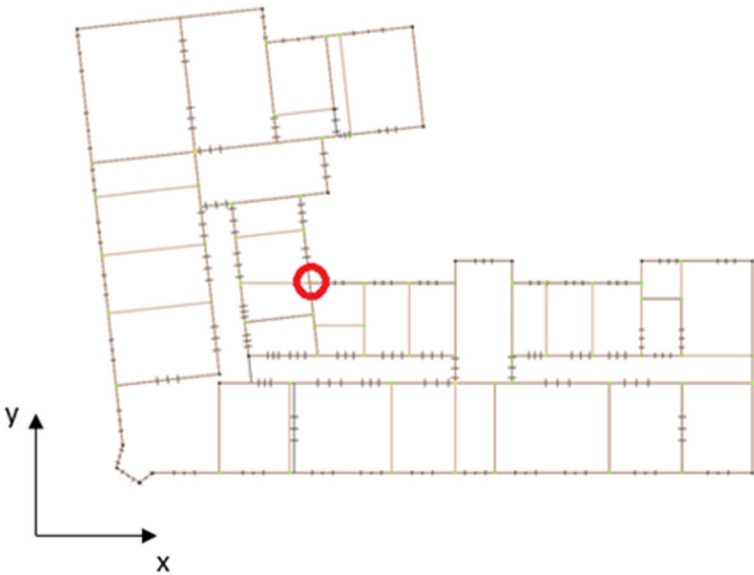

**Figure 15.** Position of the approximated center of masses on the 3rd floor–3Muri12 software.

### 4.3. Results of the Pushover Analysis

For the case study building, 24 pushover analyses were performed for x and y direction, with two load distributions in the control node at the top of the building, as shown in Figure 15. The results of the performed analysis are shown as 24 capacity curves. The shear force in the base of the structure (V in kN) is placed at the ordinate of the graph and the displacement of the control node (d in cm) is placed at the abscissa (Figure 16). The bilinearized pushover curves for the x and y direction are shown in Figures 17 and 18. The total base shear in kN is plotted on the *y* axis and the displacement of the control nodes in cm is plotted on the *x* axis.

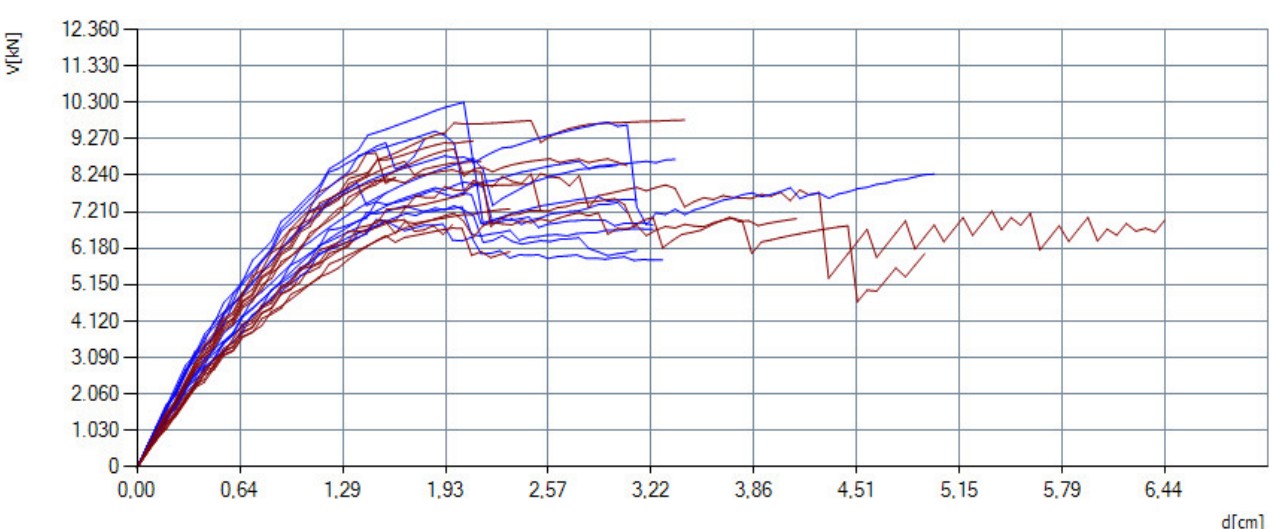

**Figure 16.** Capacity curves in the x (blue) and y (red) directions.

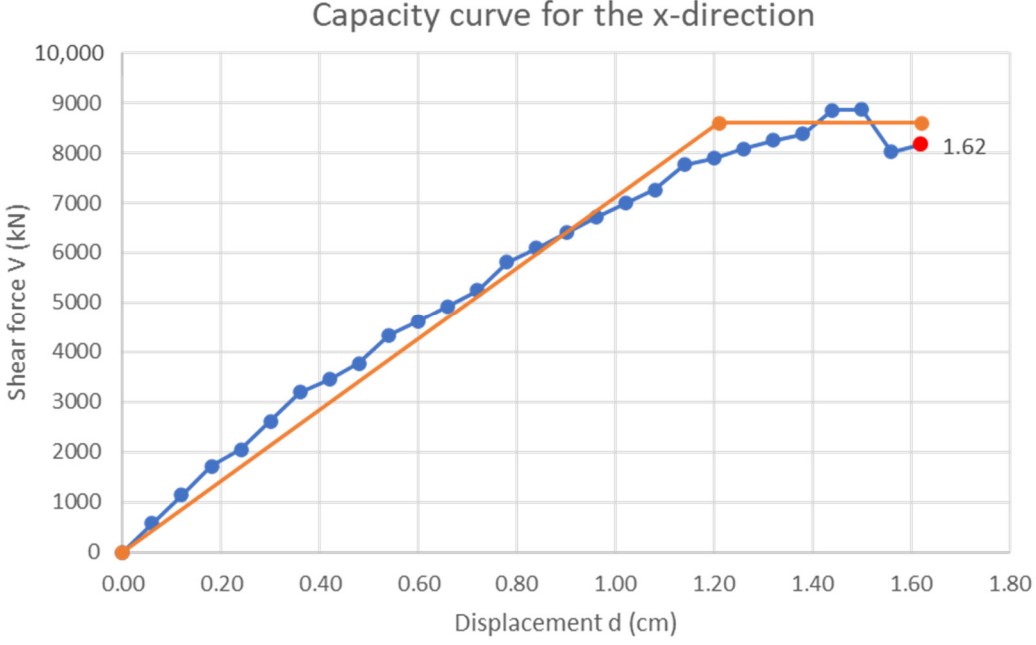

**Figure 17.** The most relevant capacity curve for x-direction.

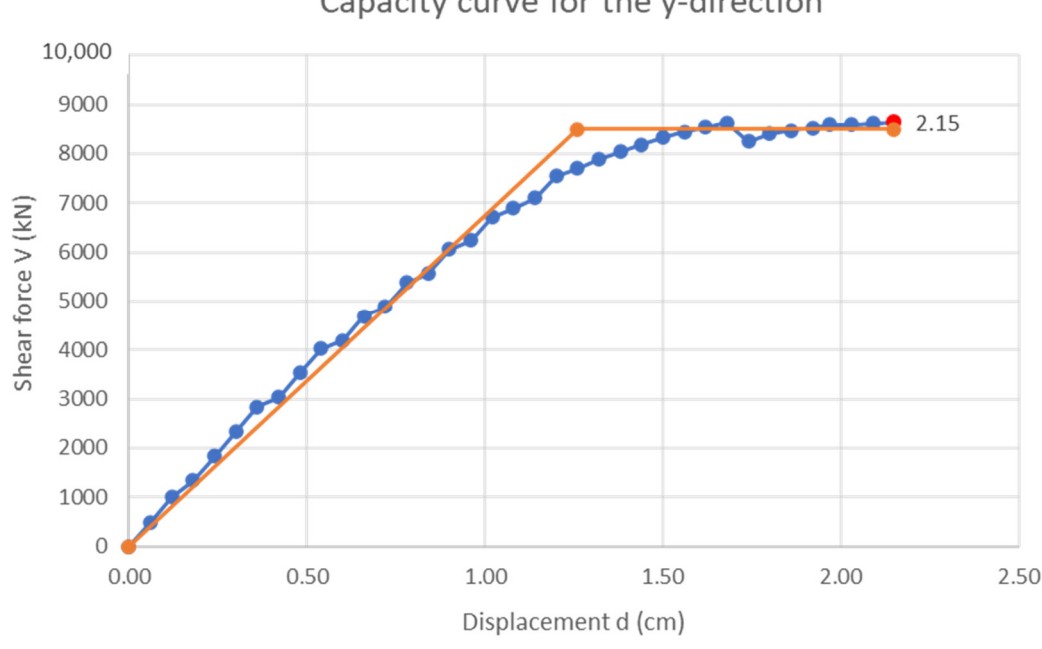

**Figure 18.** The most relevant capacity curve for y-direction.

In Figure 17, the maximum displacement achieved in the control node is 1.62 cm for the x-direction. According to Figure 18, the maximum displacement achieved in the control node is 2.15 cm for the y-direction. The total base shear equals approximately V = 8600 kN for both directions.

In Figures 19 and 20, the damaged state of the building at the last step of the pushover analysis is shown. In Figure 19, the damaged state of the building is shown for the x-direction. Most of the damage is bending damage and failure (red and pink color), with a couple of walls and spandrels being damaged in shear (beige color). A lot of tension failure may be seen as well (blue color). In Figure 20, a similar situation may be observed. However, in the y-direction, most of the building is damaged but failure has not yet occurred in any shape or form.



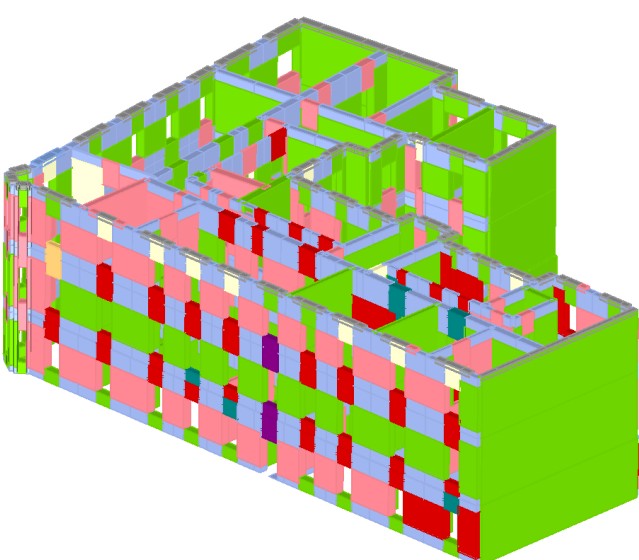

**Figure 19.** Damage at maximum displacement capacity for pushover in the x-direction.

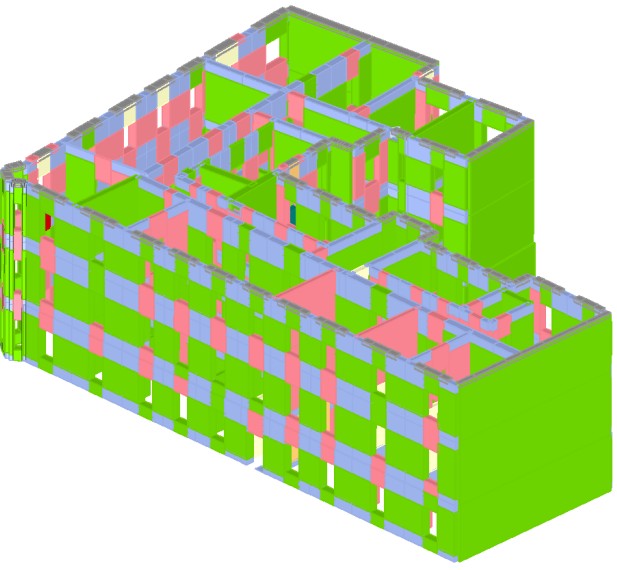

**Figure 20.** Damage at maximum displacement capacity for pushover in the y-direction.

After the structure's response, the capacity of the structure is obtained. Moreover, checks are carried out according to the basic requirements related to the state of structural damage defined by limit states. The parameters for the equivalent Single Degree of Freedom (SDOF) systems from Figures 19 and 20 are shown in Table 2. These parameters are obtained during the bilinearization of the capacity curves in the x and y directions.

The results are also given in the form of the parameter $\alpha$ shown in Table 3. The said parameter is the ratio between the limit capacity acceleration of the given building and the reference peak ground acceleration. Since all of the values are approximately equal to or lower than 0.5 it may be concluded that the building as constructed does not satisfy neither the analyses performed for the return period of 475 years nor the return period of 95 years.

Therefore, it is not surprising that for the return period of 475 years, none of the 24 analyses satisfies the limit state of significant damage (SD). The same is valid for the return period of 95 years, where none of the 24 analyses satisfy the limit state of limited damage (DL).

**Table 2.** SDOF parameters for pushover in x and y-direction.

| Mechanical Property | Value (x) | Value (y) |
|:---:|:---:|:---:|
| T (s) | 0.382 | 0.406 |
| m (kg) | 2,670,166 | 2,813,917 |
| w (kN) | 53,262 | 53,262 |
| M (kg) | 5,429,400 | 5,429,400 |
| m/M (%) | 49.18 | 51.83 |
| Γ | 1.35 | 1.33 |
| Fy (kN) | 6444 | 6324 |
| dy (cm) | 0.89 | 0.94 |
| dm (cm) | 1.2 | 1.62 |

**Table 3.** The values of parameter $\alpha$.

| Return Period | $\alpha$ (x) | $\alpha$ (y) |
|:---:|:---:|:---:|
| 475 | 0.287 | 0.319 |
| 95 | 0.547 | 0.509 |

*4.4. Results of the Out-of-Plane Bending Analysis for Walls (Local Mechanisms)*

In the pushover analysis, the 3Muri12 software does not include the bending failure of walls in the out-of-plane direction. Thus, an analysis is performed where a seismic action perpendicular to the wall is taken into account. This analysis can only be performed for the near collapse (NC) limit state with a return period of 475 years, even though the "Law on the Reconstruction of Earthquake-Damaged Buildings in the City of Zagreb, Krapina-Zagorje County and Zagreb County (NN 102/2020)" [55] states that this return period is used for the limit state of significant damage (SD). In Figure 21 the walls that did not act satisfactorily are colored red. As it may be seen, most of the walls in the case study building satisfy this analysis.

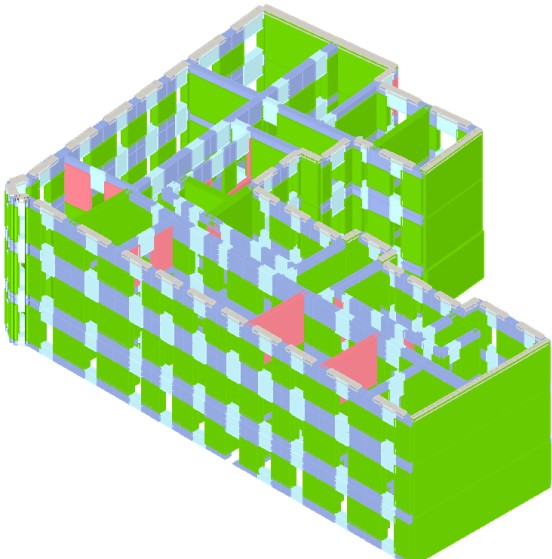

**Figure 21.** The bending out-of-plane results.

Furthermore, in the pushover analysis, the 3Muri12 software does not consider the out-of-plane loss of stability of these walls either. Throughout the pushover analysis, it is assumed that proper connections are achieved between walls and between walls and floor diaphragms. The reasoning behind such an assumption is that this is the only way to evaluate the building's global in-plane response properly. Accordingly, for the check of local

mechanisms, a special program module is imbedded into the software. It is based on linear kinematic analysis. The local mechanisms that are going to be checked are arbitrarily defined. Some of the aspects that are considered include the shape of the structure, damage found after an earthquake and common failure mechanisms found in structures of similar age and elements used. The most common reason for the appearance of such mechanisms is the poor connection details between two walls or between a wall and the floor structure of such buildings.

In the 3Muri12 software, local mechanisms are defined through three steps. First of all, a block must be defined. It is usually an entire wall or its part that is considered to be rigid. This block is then moved or tilted relative to another block that is also predefined. After that, the boundary conditions using hinges are defined. The hinge may be set as an internal or external hinge depending on the local mechanism that wants to be achieved. Finally, the direction of the seismic load is defined. This analysis can only be performed for the near collapse (NC) limit state with a return period of 475 years. Many different local failure mechanisms were checked in the numerical model, with some of these failure mechanisms shown in Figures 22–24.

In Figure 22 the first mechanism LM1 is the most common type of a local mechanism. After the Zagreb earthquake, most of the damage was detected in the form of damaged chimneys and out-of-plane failure of gable walls. As it may be seen from the quote constraint verification, this mechanism does not satisfy the analysis. In Figure 23, local mechanism LM2 shows the wall of the 2nd floor that is divided into two parts. This mechanism does not satisfy the quote constraint verification as well. In Figure 24, local mechanism LM3 shows two walls of the 2nd and 3rd floor that show an out-of-plane failure. The verification is not satisfied again.

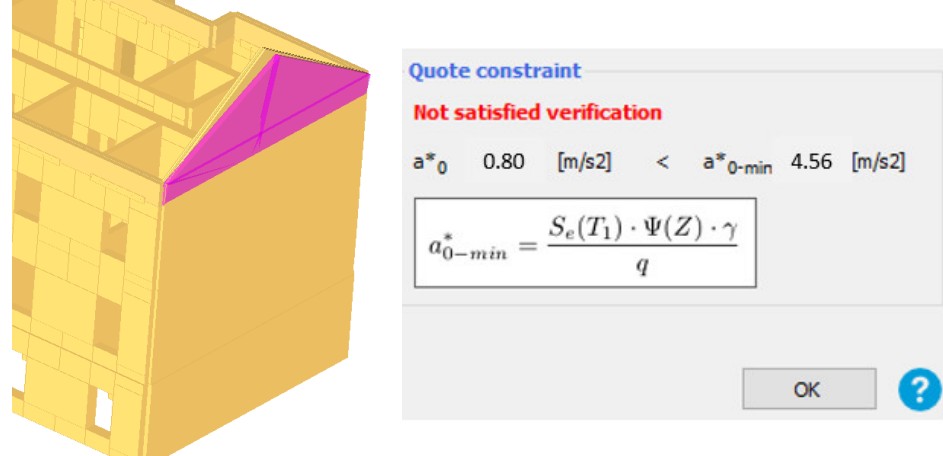

**Figure 22.** Local mechanism LM1 (Southern gable wall).

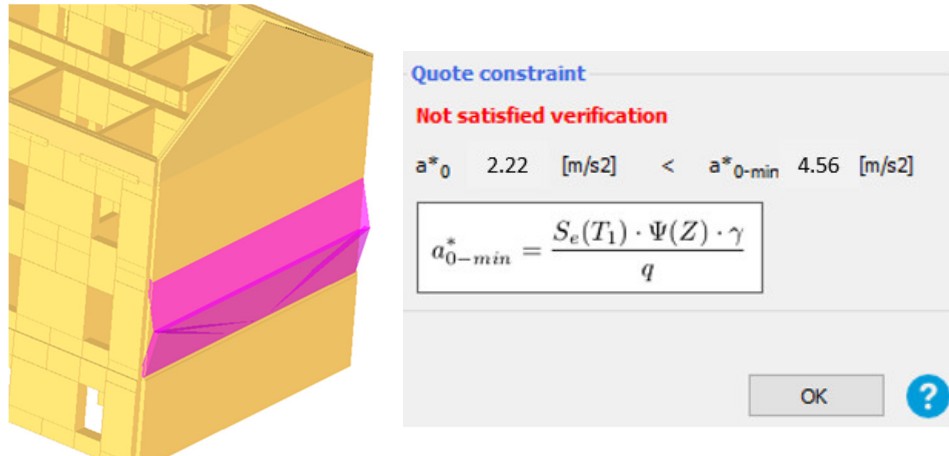

**Figure 23.** Local mechanism LM2 (Southern wall of the 2nd floor).

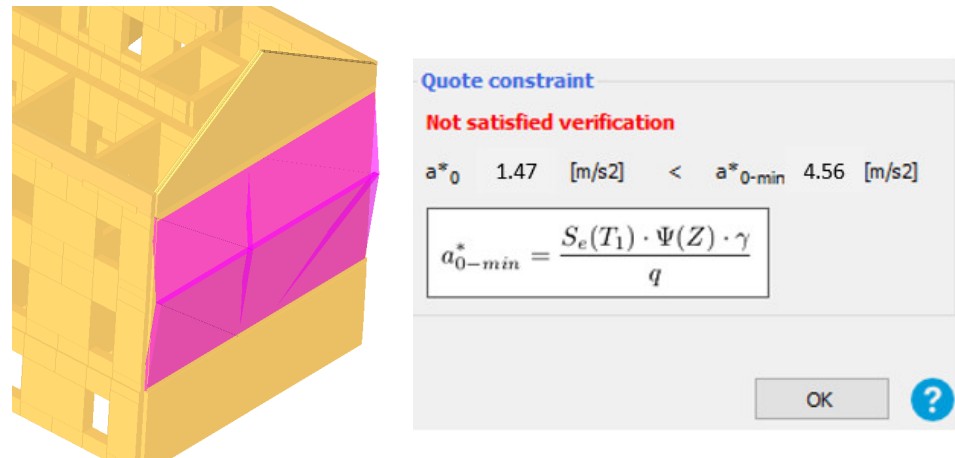

**Figure 24.** Local mechanism LM3 (Southern walls of the 2nd and 3rd floor).

*4.5. Comparison of Real-Life Damage with the Numerical Model Results*

The results obtained via the aforementioned analyses are compared to the real-life damage, as seen in Figures 25–27. Understandably, the results of the numerical analysis vary from the ones obtained in the post-earthquake assessment. However, as seen from the figures below, the damage detected in-situ is similar to the damage obtained from the software. This is perhaps the greatest value of the non-linear static pushover analysis because damage patterns may be obtained for every macro-element in every single step of the analysis.

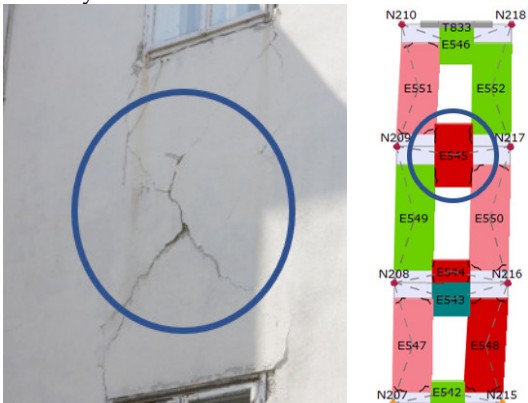

**Figure 25.** Comparison of the damaged spandrel (Eastern façade).

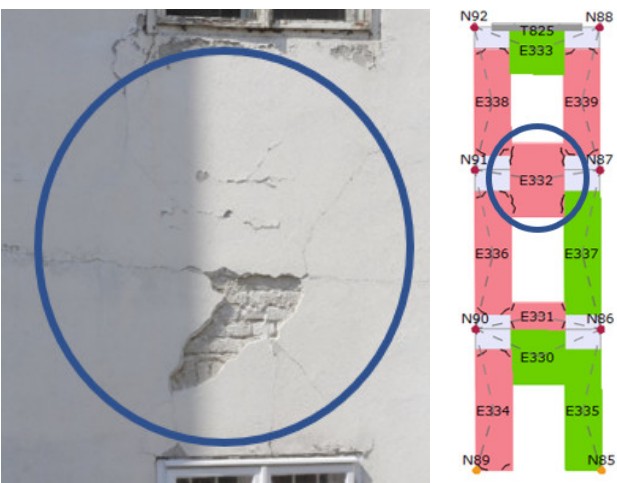

**Figure 26.** Comparison of the damaged spandrel (Southern façade).

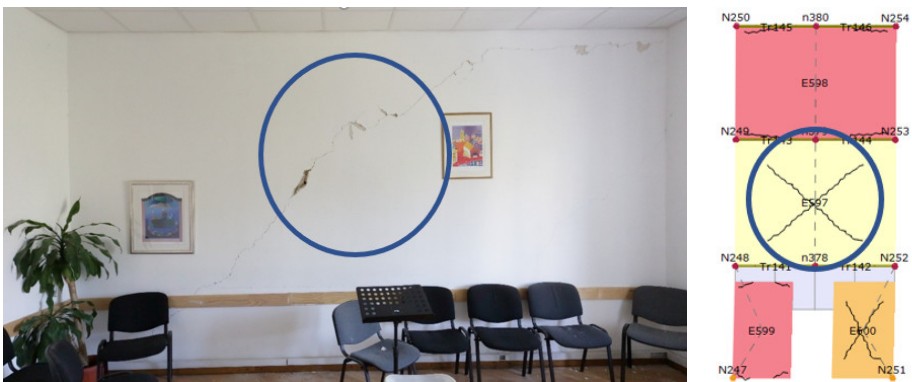

**Figure 27.** Comparison of the damaged bearing wall in the y direction.

## 5. Discussion and Conclusions

The Matica Hrvatska case study building is an old masonry building in the city center of Zagreb, Croatia. The building was damaged in the earthquakes that hit Croatia in the last year and a half. One of the most important facts about the case study building is that it is a part of an old masonry building block. It is an L-shaped corner building at the end of the building block. Since the seismic behavior of buildings connected in an aggregate is quite complex and is affected by many parameters [4,58], the observed building is usually the only one modeled. The reason for this simplification is usually the lack of information about the neighboring buildings which was also the case with the Matica Hrvatska building. Therefore, the damage obtained from the in-situ assessment cannot perfectly correspond with the damage patterns seen in the numerical model. To roughly assess the effect of adjacent buildings, the vulnerability index should be used [59,60]. Even though modeling a single unit of an aggregate is on the safe side when deformations are observed, there is a possibility of reaching the wrong conclusions about the local mechanisms. This fact may be seen from Figures 22–24 which show local mechanisms of the wall that is on the Southern side of the building. In reality there is a neighboring building on that side of the case study which prevents the occurrence of local mechanisms, at least on that side of the building. So, in the case of Matica Hrvatska case study, only the first mechanism (LM1) that was found via the 3Muri12 software corresponds to the actual damage that occurred during the earthquake. It should also be added that the non-linear static analysis used in this is paper is an approximate method and the results obtained from such a method do not perfectly correspond to the structural performance of a building under a specific ground motion [56]. In conclusion, the modeling of an entire aggregate is recommended, but the implementation of state-of-the art technologies is needed (e.g., thermography,

UAV, operational modal analysis (OMA)). It is very important to address the most common issues that may be found in aggregate buildings such as the different periods of construction, different materials used, position and type of floor structures, etc.

The second challenge while modeling such buildings are the wooden floors that are usually used. These types of floors are considered very flexible and the connections with the masonry walls are usually not well detailed [61,62]. Consequently, the behavior of the building is quite complex when using the non-linear static pushover analysis. The behavior of the building in the x and y direction is not that clear and simple as it would be with rigid diaphragms [63]. Additionally, the use of such flexible wooden floors results in a number of local mechanisms and unusual modes which prevent the full activation of the building's mass in x and y direction which is quite a problem. Using more rigid diaphragms (wooden beams with reinforced concrete slabs) enables a better connection with the walls and a more acceptable behavior of the building in general. Hence, using the 3Muri12 software and the pushover analysis in old masonry buildings with flexible wooden floors may produce ambiguous results. For the pushover analysis in 3Muri12 software, only rigid diaphragms should be used if clear seismic modes need to be obtained.

The third and final problem when conducting a pushover analysis is the building's shape [64,65]. With L-shaped buildings like the case study building, the results obtained in the x and y direction are hard to interpret. With buildings of mostly rectangular or squared shapes the behavior in the two perpendicular directions is quite clear and the displacements are adequate. The use of the pushover analysis in buildings with non-symmetrical floor plans may lead to misleading results. The main problem is the torsional effect that occurs in such buildings that results in ambiguous modes for the x and y direction in the pushover analysis. This problem may be partially solved with the use of rigid diaphragms. Moreover, a detailed analysis of the center of masses should raise the quality of the results.

With the information obtained from the visual in situ assessment and the results gathered from the pushover analysis, strengthening is needed for the case study building. The most vivid indication of this conclusion is that none of the 24 pushover analyses are satisfied. Moreover, the damage caused by recent earthquakes needs to be repaired to prevent any further progression of the damage. A new earthquake may be a real threat to the building's global seismic resistance and stability.

Since the case study building is a cultural and historical monument, as is most of the Zagreb city center, the emphasis should be on using sustainable materials and methods. Strengthening techniques such as FRP, TRM and CRM are very compatible with old masonry [66–68]. Innovative concepts and assessment methods should also be a critical part of any renovation process when talking about old masonry structures [69–71]. The visual programming for structural assessment in combination with FEM analyses [72,73] and automatic detection of cracks (with the special emphasis on out of plane mechanisms [74]) will be applied in the following stage of research by the authors.

This study provides a detailed insight into the behavior of an old masonry building during the Zagreb earthquake. The extent and distribution of damage is shown and compared to the results obtained through the pushover analysis. Furthermore, the most severe local mechanisms are detected, and critical elements are shown for earthquakes of different intensities.

**Author Contributions:** Conceptualization, I.H. and D.L.; methodology, I.H., D.L. and M.S.; software, I.H.; validation, D.L., T.K. and M.S.; formal analysis, D.L.; investigation, I.H.; resources, D.L., T.K. and M.S.; data curation, D.L.; writing—original draft preparation, I.H.; writing—review and editing, D.L., T.K. and M.S.; visualization, I.H. and M.S.; supervision, D.L.; project administration, T.K. and M.S.; funding acquisition, D.L., T.K. All authors have read and agreed to the published version of the manuscript.

**Funding:** This research was funded by the Croatian Science Foundation, grant number UIP-2019-04-3749 (ARES project—Assessment and rehabilitation of existing structures—development of contemporary methods for masonry and timber structures), project leader: Mislav Stepinac.

**Data Availability Statement:** The data presented in this study are available on request from the corresponding author. The data are not publicly available due to privacy reasons.

**Conflicts of Interest:** The authors declare no conflicts of interest.

### Appendix A

A rapid preliminary assessment is conducted as early as possible after the earthquake, bearing in mind the safety of civil engineers in the field. In Croatia, this type of assessment consisted of a quick visual inspection of individual elements of the load-bearing structure, stating the appropriate degree of damage and deciding on the classification of the building into one of six possible categories (Figure A1): U1 Usable without limitations (Green label), U2 Usable with recommendations (Green label), PN1 Temporary unusable—detailed inspection needed (Yellow label), PN2 Temporary unusable—emergency interventions needed (Yellow label), N1 Unusable due to external impacts (Red label) and N2 Unusable due to damage (Red label). The detailed explanation of each category is given as follows:

N1: Unusable—due to external risk—The building is considered dangerous due to the risk of collapse of massive parts of adjacent building (mainly gable walls and massive chimneys). It is recommended not to stay in such buildings (especially because of large number of aftershocks).

N2: Unusable—due to damage—The building has suffered significant damage to its load-bearing system, with failures of structural and non-structural elements. It is recommended not to enter or stay in the building. This does not necessarily mean that the building must be demolished—such decisions will be made at later stages.

PN1—Temporarily unusable—detailed inspection required—The building has suffered moderate damage and it is not in risk of collapse. The load-bearing capacity of the building is partially impaired. It is not recommended to stay in the building, i.e., people can stay in the building at their own risk only. Shorter stays in the building are possible, provided that the recommendations of the building expert regarding the measures to be taken and the restrictions on staying (depending on the degree of danger) are followed. The building surveyor will make recommendations to eliminate the hazard.

PN2—Temporarily unusable—short term countermeasures (urgent interventions) required—The building has suffered moderate damage and it is not in risk of collapse. However, the building cannot be used as some elements of the building are at risk of failure. The building expert determines urgent intervention measures and gives instructions to the occupants. Temporary inability to use may also be limited to some parts of the building (attic, certain story, apartment, etc.).

U1—Usable without limitations—The building is usable. The building has suffered no damage or has suffered only slight damage that cannot affect the load-bearing capacity and usability of the building.

U2—Usable with recommendation for measures to be taken—The building can be used in accordance with its intended use, with the exception of some parts of the building that pose an immediate risk. The building surveyor will make recommendations for the removal of the hazard (e.g., chimney) and recommendations to the occupants regarding temporary restriction of occupancy of certain parts of the building. Once the hazard has been removed, the building can be used without restriction.

Even more information can be found at the official website of the Croatian Centre of Earthquake Engineering: https://www.hcpi.hr/ (accessed on 1 March 2022).

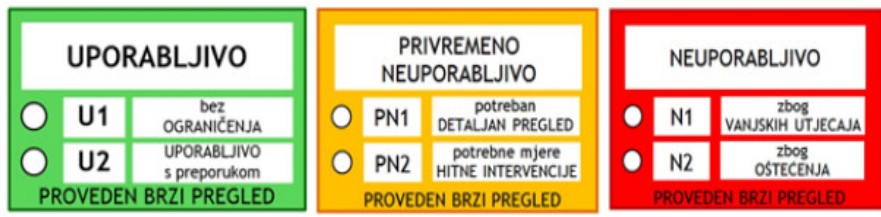

**Figure A1.** Six categories of usability divided into three original labels (in Croatian).

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
