# Peer review of "Post-Earthquake Assessment of a Historical Masonry Building after the Zagreb Earthquake—Case Study"

_buildings, doi:10.3390/buildings12030323_

Round 1
Reviewer 1 Report
The authors replied to the reviewers’ comments and have done all the modifications/corrections proposed by the reviewers. Hence, I propose the publication of the paper in the present form.
Author Response
Dear reviewer,
The whole team of authors is grateful for constructive and well-advised comments on the manuscript!
Have a nice day!
Reviewer 2 Report
The paper presents a numerical simulation of a real case study damaged after the Zagreb earthquake of March, 2020. The paper is interesting, well written and presented and it deserves of pubblication. Few comments are provided in the attached PDF file.

Author Response
Dear reviewer,
The whole team of authors is grateful for constructive and well-advised comments on the manuscript. The comments raised the quality of the paper, and we hope that we addressed all the mentioned problems.

Reviewer 3 Report
This paper presents a practical application concerning the post-earthquake assessment of historical masonry buildings after the seismic event in Zagreb 2020.
The visual inspection was fundamental to developing a nonlinear numerical model. Interesting conclusions at the end of the paper concerning the efficiency of the adopted software are presented.
I would like to propose the author to include the paper's outline at the end of the introduction section.
The description of the case study is well addressed. However, I would recommend putting a picture of the building as first figure rather than a point cloud.
The acronym UAV is called two times (line 74 and 211)
The same for NDT (75 and 211). In particular, at line 75, the authors are not consistent with the standard format (Nondestructive techniques (NDT))
Before discussing Pushover Analysis, I would suggest the authors to include the modal shapes and the participating mass factor. It is not mandatory but would increase the quality of the paper.
Would you enrich the literature with this and other journals paper?
https://doi.org/10.1016/j.jobe.2022.104182
Author Response
Dear reviewer,
Thank you very much for your comments! The changes are provided below in the word file. Good day!

Reviewer 4 Report
This research presents a discussion on the seismic behavior of buildings attached to blocks (i.e., aggregate buildings) and the urgent task of providing detailed guidance on their vulnerability assessment. In light of this, a well-developed introduction explains the state of art in the literature field and the contribution of the work reported in the paper. The authors performed seismic analysis on an important case study in Zagreb, Croatia, damaged by the earthquakes that hit the Zagreb Metropolitan area in March of 2020. At the end, interesting results are reported in detail.
Globally, the paper is well organized and well written. The quality of the figures is satisfactory.
A few issues, however, should be clarified before the paper is published:
- Please, try to improve the quality of Fig.1. It is difficult to understand the meaning of “b)” positioned at the middle of the graph. I would suggest removing it, if it is not necessary.
- In section ‘2. The case study building’ the dimensions are reported in meters and centimeters. I would suggest using all meters in order to simplify reading.
- Figures from 10 to 13 are really interesting. In my opinion, it will be useful to link damages in figures 7, 8, 9 to their location in the structure (e.g., in figures 10 to 13).
- In section ‘4. Numerical model – Pushover analysis’ the applied approach is well explained. It is clear the choice of authors in using the simplified approach proposed in the paper. However, I’d like to suggest adding some comments on considering the z-component of earthquakes for the vulnerability assessment of masonry structures. The main influence of the vertical component of a seism is already analyzed in literature (e.g., https://doi.org/10.1016/S0141-0296(02)00033-0, https://doi.org/10.1016/j.engstruct.2020.111626).
- Please, try to improve the quality of Fig.17 and 18 paying attention to the size of numbers and letters which seem too small (e.g., in Figure 16 are good for reading).
- The English language is of good quality. However, there are some typos (e.g., I would suggest using capital letters for North, South, East, West). Please, check the text.
Author Response

(The authors gave the same response as above.)
